# From Solid-State Cluster Compounds to Functional PMMA-Based Composites with UV and NIR Blocking Properties, and Tuned Hues

**DOI:** 10.3390/nano13010144

**Published:** 2022-12-28

**Authors:** Maria Amela-Cortes, Maxence Wilmet, Samuel Le Person, Soumaya Khlifi, Clément Lebastard, Yann Molard, Stéphane Cordier

**Affiliations:** 1Univ. Rennes, CNRS, ISCR, UMR6226, F-35000 Rennes, France; 2CNRS-Saint Gobain-NIMS, IRL3629, Laboratory for Innovative Key Materials ans Structures (LINK), National Institute for Materials Science (NIMS), Tsukuba 305-0044, Japan

**Keywords:** octahedral metal clusters, nanocomposites, pigment coatings, UV/NIR blockers

## Abstract

New nanocomposite materials with UV-NIR blocking properties and hues ranging from green to brown were prepared by integrating inorganic tantalum octahedral cluster building blocks prepared via solid-state chemistry in a PMMA matrix. After the synthesis by the solid-state chemical reaction of the K_4_[{Ta_6_Br^i^_12_}Br^a^_6_] ternary halide, built-up from [{Ta_6_Br^i^_12_}Br^a^_6_]^4−^ anionic building blocks, and potassium cations, the potassium cations were replaced by functional organic cations (Kat^+^) bearing a methacrylate function. The resulting intermediate, (Kat)_2_[{Ta_6_Br^i^_12_}Br^a^_6_], was then incorporated homogeneously by copolymerization with MMA into transparent PMMA matrices to form a brown transparent hybrid composite Ta@PMMA_brown_. The color of the composites was tuned by controlling the charge and consequently the oxidation state of the cluster building block. Ta@PMMA_green_ was obtained through the two-electron reduction of the [{Ta_6_Br^i^_12_}Br^a^_6_]^2−^ building blocks from Ta@PMMA_brown_ in solution. Indeed, the control of the oxidation state of the Ta_6_ cluster inorganic building blocks occurred inside the copolymer, which not only allowed the tuning of the optical properties of the composite in the visible region but also allowed the tuning of its UV and NIR blocking properties.

## 1. Introduction

The conception of new advanced materials for energy saving for eco-friendly building design is a subject attracting great interest. Indeed, the increase in energy consumption by excessive heating and/or air conditioning leads to the use of fossil fuels and thus to high emission of carbon dioxide. In this context, the development of low-cost coatings, transparent in the visible range of wavelength and allowing the control of the sunlight and heat passing through seems very promising [1,2]. Hence, on the one hand, for glazing applications, it is important to find ultra-violet (UV) blockers stopping radiations (300−400 nm) that degrade organic matter such as polymers along with the formation of volatile organic compounds [3,4,5]. On the other hand, near-infrared (NIR) rays (700−3000 nm) are responsible for heat radiation in buildings since they can be absorbed by the transparent materials used for glazing, for instance, and emitted at a higher wavelength (8 µm–12 µm). Efficient solar control windows will prevent the use of air conditioning systems and consequently, will contribute to reducing the carbon impact of buildings. Since conventional glasses and polymers used in buildings do not reflect or absorb the NIR radiations, solar control coatings are commonly used for such applications. They are made of complex metal- and oxide-based stacks obtained by physical vapor deposition [6]. However, the technology is quite expensive and reveals some limitations in terms of performance. In the field of energy-saving applications, many inorganic materials are known and used as NIR reflectors as for instance TiO_2_, and BiVO_4_ for cool roofing applications [7]. In the frame of aesthetic purpose, many investigations are devoted to the search for pigments combining NIR reflectance with coloring abilities. Inorganic pigments showing NIR reflectance properties have been already classified and grouped by color families [8]. 

For glass coating applications, several publications dealing with the use of selective inorganic absorbers dispersed in organic polymer matrices. As promising NIR blockers for incorporation in composite materials, let us cite LaB_6_, indium oxide, hexagonal tungsten bronzes, indium-doped CuS nanocrystals and g-C_3_N_4_@CsWO_3_ heterostructures [9,10,11,12,13,14,15]. However, until now, color adjustment is not yet a real objective.

Recently, Nb and Ta cluster-based compounds have emerged as a new class of inorganic UV/ NIR absorbers. Composites were obtained by direct integration of colloidal solutions, containing dispersed K_4_[{Nb_6−x_Ta_x_X^i^_12_}X^a^_6_], in polyethyleneglycol-silicium dioxide (PEG-SiO_2_) and polyvinylpyrrolidone (PVP) matrices. The use of the latter matrices shows significant drawbacks which are prohibitive for potential applications: (i) after aging, the oxidation of building blocks is observed in PEG-SiO_2_ and (ii) PVP is a water-soluble matrix [16,17]. 

In this work, we present how to use solid-state cluster compounds to prepare robust and functional polymethylmethacrylate (PMMA) based composites exhibiting UV-NIR blocking properties and tuning hues from green to brown. Inorganic cluster halides, chalcogenides and oxides are prepared at high temperatures by solid-state chemistry technics. Their structures are characterized by well-defined aggregates of metal atoms bonded to non-metal atoms as defined by F.A. Cotton [18]. Among the family of inorganic clusters, those based on octahedrons are particularly attractive in terms of structural diversities, physical properties and potential applications. Octahedral clusters are stabilized by ligands to form either face-capped [{M_6_Q^i^_8_}L^a^_6_]^n−^ cluster building blocks (M: Mo, Re, W; Q: halogen or chalcogen and L: halogen) or [{M_6_X^i^_12_}X^a^_6_]^n−^ edge-bridged ones (M: Ta, Nb; X = Cl, Br). Face-caped cluster building blocks have been the focus of attention because of their emission in the deep red region [19,20] making them, after prior functionalization and shaping of solid-state compounds, good candidates for optoelectronic devices, biotechnology, photovoltaic cells and photocatalysis [21,22,23,24,25,26,27]. In this work, we will focus on using K_4_[{Ta_6_Br^i^_12_}Br^a^_6_] to design functional hybrid composites. K_4_[{Ta_6_Br^i^_12_}Br^a^_6_] is a solid-state compound that crystallizes in the K_4_Nb_6_Cl_18_-type structure (space group *C2/m*, No. 12; a = 10.46(1) Å, b = 17.19(2) Å, c = 9.99(1) Å, β = 114.89(1)°, and V = 1629.1(3) Å 3; Z = 2) (Figure 1) [28]. The structure is built up from [{Ta_6_Br^i^_12_}Br^a^_6_]^4−^ building blocks whose charge is counterbalanced by potassium cations [29]. This solid-state compound that can be viewed as a functional inorganic salt has been poorly studied until very recently for the design of functional materials although it exhibits very rich electronic and absorption properties [30,31,32,33,34]. The pioneering work of Chabrié and Harned showed that solid-state compounds based on [{Ta_6_Br^i^_12_}Br^a^_6_]^n−^ cluster building blocks exhibit a strong coloring power when they are put in solution [35,36,37]. Tantalum bromides containing cluster building blocks such as [{Ta_6_Br^i^_12_}Br^a^_6_]^n−^ (n = 4, 3, 2) or [{Ta_6_Br^i^_12_}(H_2_O)^a^_6_]^m+^ made of {Ta_6_Br^i^_12_}^m+^ (m = 2, 3, 4) cluster core have found applications in catalysis, photocatalysis, X-ray crystallography, radiographic contrast agents and as a green-emerald pigment [29,38,39,40,41,42]. 

The aim of this work is (i) to fabricate colored homogeneous and transparent free-standing PMMA films by embedding inorganic Ta_6_ cluster building blocks as pigments capable to absorb UV radiation and the most energetic short wavelength NIR radiation (700−1100 nm) and (ii) to demonstrate that control of hues (brown-green) can be stabilized in the polymer matrix by modulating the oxidation state of the inorganic Ta_6_ clusters. In this work, we use a covalent/electrostatic approach in order to immobilize the Ta_6_ units in the polymer backbone. Thus, after oxidation in a solution of the K_4_[{Ta_6_Br^i^_12_}Br^a^_6_] precursor, the resulting oxidized [{Ta_6_Br^i^_12_}Br^a^_6_]^2−^ cluster building block is efficiently paired with two organic cations (Kat) bearing a methacrylate unit to form a stable (Kat)_2_[{Ta_6_Br^i^_12_}Br^a^_6_] intermediate. Thanks to Kat^+^ cations, (Kat)_2_[{Ta_6_Br^i^_12_}Br^a^_6_] is able to copolymerize, by a solvent-free method, with the methylmethacrylate (MMA) monomer to form brown composites Ta@PMMA_brown_. The [{Ta_6_Br^i^_12_}Br^a^_6_]^2−^ cluster building block interacts by electrostatic interactions with the Kat^+^ cations incorporated in the MMA-based chains of macromolecules. After the dissolution of Ta@PMMA_brown_ composites in dichloromethane, [{Ta_6_Br^i^_12_}Br^a^_6_]^2−^ building blocks can be reduced by the addition of dimethylaminopyridine (DMAP) to form [{Ta_6_Br^i^_12_}Br^a^_6_]^4−^ within the PMMA matrix. After evaporation of the solvent, green composite Ta@PMMA_green_ is recovered. Both Ta@PMMA_brown_ and Ta@PMMA_green_ films exhibit UV and NIR blocking properties with some shifts of the maxima of absorption depending on the charge of the cluster building block, i.e., [{Ta_6_Br^i^_12_}Br^a^_6_]^2−^ or [{Ta_6_Br^i^_12_}Br^a^_6_]^4−^.

## 2. Materials and Methods

K_4_[{Ta_6_Br^i^_12_}Br^a^_6_] was obtained by a reported procedure [28,43]. Briefly, K_4_[{Ta_6_Br^i^_12_}Br^a^_6_] is synthesized by the reduction of the tantalum bromide (Alfa Aesar, Heysham, Lancashire, UK, 99.9%, CAS 13451-11-1) by an excess of tantalum powder (Alfa Aesar, 99.97%, CAS 744-25-7) at 700 °C in the presence of KBr (ACROS, Thermo Fisher Scientific, Geel, Belgium, 99%, CAS 7758-02-3). The reaction takes place in a silica container. The fully optimized experimental procedure is reported in [28].

Dodecyl(11-(methacryloyloxy)undecyl)dimethylammonium Bromide (Kat-Br) was synthesized following two steps described procedure [44,45,46]. Briefly, in the first step methacryloyl chloride (Aldrich, Darmstadt, Germany, 97%, CAS 920-46-7) reacted with 11-bromoundecanol (Aldrich, 98% CAS 1611-56-9), to form the 11-undecenylmethacrylate that was used in a second step to alkylate the NN-dimethyldodecylamine (Aldrich, 97%, CAS 112-18-5).

Previous to polymerization, Methyl methacrylate (MMA (Aldrich, 99%, CAS 80-62-6) was distilled and Azobisisobutyronitrile (AIBN Aldrich, 98%, CAS 78-67-1) was purified by recrystallization in diethylether.

Dimethylaminopyridine (DMAP, Aldrich, 99%, CAS 1122-58-3) was used as received.

### 2.1. Synthesis

#### 2.1.1. Synthesis of the (Kat)_2_[{Ta_6_Br^i^_12_}Br^a^_6_] Intermediate by Metathesis Reaction

A similar metathesis reaction procedure as that reported for the preparation of ((C_4_H_9_)_4_N)_2_[{Ta_6_Br^i^_12_}Br^a^_6_] from K_4_[{Ta_6_Br^i^_12_}Br^a^_6_] was used to prepare (Kat)_2_[{Ta_6_Br^i^_12_}Br^a^_6_] [28]. Briefly, a solution containing K_4_[{Ta_6_Br^i^_12_}Br^a^_6_] (0.06 g, 0.015 mmol) was prepared by dissolution of K_4_[{Ta_6_Br^i^_12_}Br^a^_6_] solid-state precursor in 5 mL of acetone. The solution became brown due to the two-electron oxidation of the [{Ta_6_Br^i^_12_}Br^a^_6_]^4−^ the building block to form [{Ta_6_Br^i^_12_}Br^a^_6_]^2−^. After that, the solution was filtered over Celite^®^. Kat-Br (2 equivalents, 0.016 g, 0.03 mmol) was added to the solution and the mixture was stirred for 24 h at room temperature. After filtration of the KBr formed, the solvent was evaporated and the product was dried under vacuum and obtained as a dark brown oil. 

^1^H-NMR (400 MHz, CDCl_3_) δ ppm: 6.11 (d, 2H, C*H*H=C), 5.57 (d, 2H, CH*H*=C), 4.15 (t, 4H, –CH_2_–O), 3.51 (br, 8H, –CH_2_–N^+^), 3.38 (s, 6H, CH_3_–N^+^), 1.94 (s, 6H,–CH_3_–C), 1.7–1.67 (m, 8H, –(CH_2_CH_2_)_2_N, 1.66 (m, 4H, –CH_2_–CH_2_–O), 1.33–1.26 (m, 64H, –CH_2_), 0.86 (t, 6H, –CH_3_).d: doublet, t: triplet, br: broad signal, s: singlet, m: multiplet.EDAX: no potassium Ta 27%, Br 73% (Theo: 25%, Br 75%).

#### 2.1.2. Polymerization with Methyl Methacrylate

The polymerization was performed in bulk. For 1 g of polymer, the polymerizable (Kat)_2_[{Ta_6_Br^i^_12_}Br^a^_6_] cluster precursor was dissolved in freshly distilled methylmethacrylate (from 1 wt% to 3 wt%). After the addition of 0.5 wt% of the radical initiator AIBN, the solutions were placed at 70 °C for 1 h and then the temperature was lowered to 60 °C for 24 h to complete polymerization. Brown pellets of the hybrid polymers were then obtained. Hybrid polymers will be denoted as Ta_x@PMMA_brown_ (x = 1, 2, 3 for Ta content of 1, 2 and 3 wt%, respectively). ^1^H-NMR spectra of dissolved pellets in CDCl_3_ are provided in the Appendix A.

#### 2.1.3. Ta_x@PMMA_brown_ Composites Film Deposition

Ta_x@PMMA_brown_ composite films were obtained by solvent casting method by dissolving 0.5 g of each polymer in 20 mL of HPLC grade dichloromethane and pouring the solutions into glass Petri dishes from which the solvent was slowly evaporated. Films of 0.25 mm thickness were obtained.

#### 2.1.4. Reduction of Ta_x@PMMA_brown_ Polymer Composites

Ta_x@PMMA_brown_ composite pellets (0.5 g) obtained by polymerization in the oxidized form were dissolved in 20 mL of HPLC-grade dichloromethane. To that, aliquots of a stock solution of DMAP (10 mg/mL) corresponding to two equivalents of the Ta_6_ complex building block in the polymer were added and the mixture was stirred for 24 h at room temperature. Amounts of Ta_6_ cluster and DMAP are provided in Table 1. The color of the solution changed from brown to emerald green, which is the typical color of species based on [{Ta_6_Br^i^_12_}^2+^ cluster core with a valence electron concentration (VEC) of 16. The green solutions were poured into glass Petri dishes and the solvent was allowed to slowly evaporate. Films denoted as Ta_x@PMMA_green_ of 0.25 mm thickness were obtained.

### 2.2. Instrumentation

#### 2.2.1. NMR Experiments

^1^H-NMR was realized on a Bruker Avance III 400 MHz NMR spectrometer (Brucker, Billerica, MA, USA). Approximately 5 mg of the compound was dissolved in 700 μL of deuterated solvent. Spectra were analyzed on the MestReNova, 14 software (Mestrelab research S. L., Santiago de Compostela, Spain). 

#### 2.2.2. Energy Dispersive Scattering (EDS)

EDS analyses were realized in the CMEBA-SCanMAT platform. The chemical analyses of the heavy elements Ta and Br of the films Ta3@PMMA_brown_ and Ta3@PMMA_green_ were carried out using an SEM 7100 F Jeol (JEOL Ltd., Tokyo, Japan) equipped with a detector EDS Oxford X-Max (Oxford Instruments PLC, Abingdon-on-Thames, UK) for elemental analysis. Before analyses, the samples were metalized with gold and palladium. For each sample, two zones of 100 × 100 μm^2^ were arbitrarily selected. The chemical analyses were then performed for 15 points homogeneously distributed within each of the selected zones. The size of the irradiated area for each point was roughly equal to 1 × 1 μm^2^. The time of irradiation was 15 s and the acceleration voltage was 20 kV.

#### 2.2.3. Infrared (IR) Spectroscopy

K_4_[{Ta_6_Br^i^_12_}Br^a^_6_], (Kat)_2_[{Ta_6_Br^i^_12_}Br^a^_6_] Ta_6_ cluster complex, PMMA and composite films were all analyzed using Universal Attenuated Total Reflectance Accessory FT-IR Spectroscopy (Bruker, vertex 70, Billerica, MA, USA). Mid-infrared spectra were recorded in the range from 500 to 5000 cm^−1^ and far-infrared spectra were analyzed from 50 to 700 cm^−1^. All spectra were analyzed with OPUS 7 software (Bruker, Billerica, MA, USA).

#### 2.2.4. Thermal Analysis

Differential Scanning Calorimetry (DSC) measurements were carried out in a DSC25 (Waters, TA Instruments Ltd., New Castle, DE, USA) calibrated using purified indium (99.9%) as the standard reference material. Samples (4–5 mg) were cut off and sealed in an aluminum pan. They were heated at a constant rate of 10 °C min^−1^, using a dry atmosphere of argon as a carrier gas, in a temperature range of 30 to 150 °C. The glass transition temperature of neat PMMA and Ta_x@PMMA copolymers was obtained from the midpoint transition of the second heating curve using TRIOS 5.4 Software (Waters, TA Instruments, New Castle, DE, USA). 

Thermogravimetric analyses (TGA) measurements were carried out in a TGA8000 (Perkin Elmer, Waltham, MA, USA). Samples (~5 mg) were placed in a ceramic crucible and heated from 50 to 500 °C at a constant rate of 10 °C min^−1^. The decomposition temperatures were determined using Pyris 11 software ((Perkin Elmer, Waltham, MA, USA). 

#### 2.2.5. UV-Vis-NIR Experiments

Optical transmittance of Ta_x@PMMA films was measured in a Perkin-Lambda 1050 (Perkin Elmer, Waltham, MA, USA) in the range 250–2500 nm. Films were measured directly. The CIE colorimetric system was used to determine the L, a* and b* color coordinates following reported procedures (CIE: Comission Internationale de l’Eclairage, 1931, with 10° Standard Observer from 1964 and D5 source) [17]. The values T_vis_ (visible transmittance) and T_sol_ (solar transmittance) were calculated from reported procedures [47,48]. 

## 3. Results

### 3.1. Synthesis of Functional Ta_6_ Cluster Precursors and Ta_x@PMMA Composites

The strategy used here consists of the exchange in a solution of the potassium cations of the ternary halide K_4_[{Ta_6_Br^i^_12_}Br^a^_6_] by functional organic cations (Kat^+^) bearing a methacrylate function in the terminal position (Figure 2a). It is important to note that K_4_[{Ta_6_Br^i^_12_}Br^a^_6_] fully dissolved and that in solution, the [{Ta_6_Br^i^_12_}Br^a^_6_]^4−^ cluster building blocks do not form aggregates. Recently, it was evidenced that the dissolution of green powdered K_4_[{Ta_6_Br^i^_12_}Br^a^_6_] in acetone under stirring in air leads to dark brown solutions. The color of the solution that goes from green-emerald to brown is directly related to the charge and consequently to the oxidation state of the cluster building blocks [28,43,49,50,51]. Indeed, Ta_6_ cluster-based building blocks are very easily and reversibly oxidized from a VEC (Valence electron count) of 16 to a VEC of 14, allowing the modulation of the optical properties. Therefore, after the addition of two equivalents of Kat-Br to an acetone solution of [{Ta_6_Br^i^_12_}Br^a^_6_]^2−^ cluster building blocks, and subsequent purifications (i.e., elimination by filtration of the KBr salt formed), the (Kat)_2_[{Ta_6_Br^i^_12_}Br^a^_6_] intermediate was obtained as a brown viscous oil. The ammonium salt (Kat-Br) was synthesized according to previously described methods [44,45,46]. 

Afterward, the organic cations of (Kat)_2_[{Ta_6_Br^i^_12_}Br^a^_6_], react with MMA co-monomers following a solvent-free method. The organic cations are incorporated covalently within the polymer chains while interact electrostatically with [{Ta_6_Br^i^_12_}Br^a^_6_]^2−^, anionic metal cluster building blocks. These interactions enable: (i) to improve the dispersion of the cluster building blocks into the organic polymer, (ii) to form stable composites for which the cluster building blocks will not aggregate and (iii) to avoid leakage of the cluster building blocks (Figure 2b). 

This strategy, derived from that used for the design of nanocomposites integrating face-capped clusters, was successfully adapted herein for the first time for edge-bridged tantalum metal atom-based clusters. The polymerization was performed in the bulk by mixing 1, 2 and 3 wt% of (Kat)_2_[{Ta_6_Br^i^_12_}Br^a^_6_] and neat MMA in two steps. First, a pre-polymer was formed at 70 °C for one hour followed by a treatment at 60 °C for 24 h to complete the reaction.

Brown homogeneous pellets denominated Ta_1@PMMA_brown_, Ta_2@PMMA_brown_ and Ta_3@PMMA_brown_ (containing 1, 2 and 3 wt% Ta_6_ cluster, respectively) were obtained (Figure 2c). Pellets with a higher ratio of (Kat)_2_[{Ta_6_Br^i^_12_}Br^a^_6_] were tempted but resulted in non-homogeneous composites since their solubility in the matrix became very difficult. After polymerization, the pellets were dissolved in chloroform, precipitated from methanol and filtered out to remove non-reacted monomers. The absence of brown color in the filtrate indicated that the (Kat)_2_[{Ta_6_Br^i^_12_}Br^a^_6_] cluster intermediate is fully bounded onto the PMMA matrix and that the strength of electrostatic interactions between cationic PMMA-based matrix and [{Ta_6_Br^i^_12_}Br^a^_6_]^2−^ cluster building blocks prevents their leakage. The ^1^H-NMR of the three Ta_x@PMMA_brown_ compositions showed the disappearance of the signals of the double bond of the methacrylate group at 6.2 and 5.5 ppm and the absence of the singlet at 1.9 ppm corresponding to the methyl protons of the methacrylic units of Kat^+^ and MMA. It has to be noted that the amount of cluster building blocks is too low to show the corresponding signals (see Appendix A).

### 3.2. Modulation of Oxidation States of Ta_6_ Cluster Building Blocks in PMMA Matrix and Characterization of Thin Films

In this work, we demonstrate that the brown composites Ta_x@PMMA_brown_ can be chemically reduced in solution to afford the corresponding green composites and thus, the optical behavior of the composites can be modulated.

As stressed above, the Ta_6_ cluster based building blocks with VEC = 14 can be reversibly reduced to VEC = 16. Organic compounds have been preferred to metallic reducing agents to avoid possible absorption and to improve solubility in the polymer matrix in common organic solvents. Strong reducing agents that could attempt the integrity of the polymer backbone have also been avoided. Thus, several organic compounds such as hydrazine, urea, pyrrole and 4,4-dimethylpyridine (DMAP) have been investigated. Among them, DMAP revealed the best performance in terms of the time of reduction, the color of the solution and the solubility of species. Thus, oxidized Ta_6_-cluster building blocks (VEC = 14) could undergo complete reduction (VEC = 16), by using two equivalents of DMAP per cluster in dichloromethane (Figure 2d,e). The Ta_6_ building block solution changes from brown-orange to green after 24 h of reaction. DMAP is an electron donor that effectively reduces Ta_6_ cluster building block whilst converted into positively charged pyridinium species. To assess the formation of a pyridinium ring Fourier Transformed Infra-red (FT-IR) and ^1^H-RMN have been conducted before and after the reaction. Thus, to a solution of brown (TBA)_2_[{Ta_6_Br^i^_12_}Br^a^_6_]^2−^ cluster units in dichloromethane, two equivalents of DMAP were added and the reaction was allowed to proceed for 24h (see Appendix A). The ^1^H-NMR spectrum shows a shift of the aromatic and methyl protons indicating a redistribution of the charge density. The absence of methylene signal at 6.23 ppm, excludes the formation of bis(pyridinium)methane dicationic species [52]. Thus, two pyridinium cations should counterbalance the 4- charge of the Ta_6_ cluster building block along with two TBA cations. By FT-IR spectroscopy, we observed the disappearance of the stretching bands at 1537, and 1518 cm^−1^ while that at 1595 cm^−1^ becomes weaker. Those bands are attributed to the vibrations of C=C and C=N of the pyridine ring. The appearance of stretching bands at 1643 and 1560 cm^−1^ of the C=C and C=N vibrations as well as the band at 1260 cm^−1^ corresponding to the stretching N−H vibration is consistent with the bands reported in the literature for pyridinium rings [53,54]. 

Ta_x@PMMA_brown_ brown films were obtained by solvent casting from a solution containing 0.5 g of the corresponding pellet in 10 mL of dichloromethane. Reduced Ta_x@PMMA_green_ films were realized by dissolving 0.5 g of the corresponding pellet and two equivalents of DMAP per cluster compound in 10 mL of dichloromethane. After 24 h, the solution was poured into a Petri dish and the solvent was slowly evaporated. Transparent films 0.25 mm thick were obtained (Figure 2e). Thus, the covalent embedding of polymerizable Ta_6_ building blocks allows the processing of nanocomposites without any loss of cluster building blocks. The good dispersion of Ta_6_ building blocks within the polymer and thus the homogeneity of films has been determined by Scanning Electron Microscopy (SEM) combined with Energy Dispersive Scattering (EDS) for chemical analysis of heavy atoms Ta and Br for Ta_3@PMMA_green_ and Ta_3@PMMA_brown_. Chemical analyses for both films were performed for 15 points homogeneously distributed within two zones arbitrarily selected of 100 × 100 μm^2^. For Ta_3@PMMA_brown_, all the analyzed points show a similar composition for Ta and Br of around 25% and 75%, respectively, which is close to the theoretical one for the [{Ta_6_Br^i^_12_}Br^a^_6_] cluster unit (Ta: 25%; Br: 75%). For Ta_3@PMMA_green_, all the analyzed points show a similar composition for Ta and Br close to 15% and 85%, respectively. The deviation to the 25%:75% theoretical ratio may be explained by the fact that the reduced [{Ta_6_Br^i^_12_}Br^a^_6_]^4−^ cluster units are partially decomposed under the electron beam. However, the fact that for each point the composition is the same with the error bar reflects the homogeneous distribution of the [{Ta_6_Br^i^_12_}Br^a^_6_]^4−^ cluster units with the polymer.

The thermal behavior of oxidized and reduced Ta_x@PMMA composites was studied by differential scanning calorimetry (DSC) and thermogravimetric analysis (TGA). The influence of the embedment of inorganic cluster units into PMMA on the glass transition temperature *T*_g_ has been investigated. The *T*_g_ decreases from 114 °C for neat PMMA, obtained under the same conditions, to around 85 °C after the introduction of (Kat)_2_[{Ta_6_Br^i^_12_}Br^a^_6_] cluster precursor. The addition of two equivalents of DMAP induces a slight decrease in the *T*_g_ value of 6 °C for Ta_1@PMMA_green_ while for 2 and 3 wt% loading the *T*_g_ stabilizes at higher temperatures (93 and 97 °C, respectively) compared to brown films at the same loading ratio (Table 1, Figure 3a, Appendix A). The thermal stability of neat PMMA, brown and green nanocomposites was determined by thermogravimetric analysis (TGA) (Figure 3b–d). The introduction of (Kat)_2_[{Ta_6_Br^i^_12_}Br^a^_6_] cluster precursor in the most oxidized form (brown films) has little effect on the decomposition temperature of neat PMMA. Brown film at 3w% cluster content induces a slight decrease in the decomposition temperature of PMMA from 400 °C to 390 °C while for 1 wt% and 2 wt% a slight increase to 407 and 415 °C, respectively (Figure 3b,d). This behavior is expected since the amount of loaded inorganic cluster units is small and corresponds well to that observed for other cluster compounds of Mo, W and Re embedded at 1 wt% into PMMA [44,45]. For the reduced green films containing the same cluster units content, the decomposition temperature is stabilized to 420 °C for 1 wt% and up to 440 °C for 2 and 3 wt% loading (Figure 3c). Figure 3b shows a comparison between the behavior of neat PMMA and the brown and green composites with 3 wt% loading and a clear stabilization of the thermal decomposition can be observed. Thus, the cluster building blocks content is not the only issue to take into account but also the introduction of more amount of DMAP into the polymer at 3 wt% has to be considered. 

Infrared spectra of Ta_x@PMMA nanocomposite films were recorded in the mid-IR region (4000−500 cm^−1^) and in the far-IR region (700−50 cm^−1^) in order to assess if the introduction of Ta_6_ cluster compounds has an influence on the structure of the PMMA matrix. In the mid-IR region, (Appendix A) the characteristic absorption bands of PMMA are observed for all oxidized and reduced composite samples and thus, any influence of the cluster compound oxidation state is detected. The band observed at 3100–2900 cm^−1^ corresponds to the stretching vibration of –CH_3_, −CH_2_ groups. At 1730 cm^−1^ the intense band is attributed to the carbonyl C=O stretching band and at 1250 cm^−1^ appears the –C−O stretching band of the ester groups. In the three spectra of the reduced green composite films appears a band at 1598 cm^−1^ that can be attributed to the C=N stretching vibrations of the pyridinium ring.

Several far-FTIR studies have been reported in the literature for compounds containing {M_6_X_12_} cluster core-based building blocks (M = Nb, Ta; X = Cl, Br) in the three different oxidation states [55,56,57]. In the case of [{Ta_6_Br^i^_12_}Br^a^_6_]^2−^ six fundamental infrared active modes were assigned, four corresponding to the {Ta_6_Br_12_}^n+^ cluster core, one arises from the stretching of Ta-Br^a^ bonds and one from bending of Ta-Br^a^ bonds [28]. Experimental data of cluster building blocks and polymers are compared to those reported in the literature (see Appendix A). The one reported at 98 cm^−1^ is not visible but a large band is observed between 90−80 cm^−1^. The far-FTIR spectra of neat PMMA showed very broad absorptions that are in agreement with those reported in the literature (Appendix A)) [58]. The absorption bands above 500 cm^−1^ and at 480 cm^−1^ have been attributed to the out-of-plane bending of C−C−C bonds from the PMMA backbone. The strong absorption at 360 cm^−1^ was assigned to the C−C−C bending/twisting whereas the broad less intense band at around 200−215 cm^−1^ was attributed to a weak internal mode of the polymer. The spectra of polymer composites present very broad bands similar to that of the PMMA matrix except for the reduced Ta_3@PMMAgreen that shows a very intense absorption at 470 cm^−1^. The cluster building block absorption bands are not evidenced because of their low concentration within the host.

### 3.3. Optical Properties of Ta_6_@PMMA Nanocomposite Films in the Oxidized and Reduced Form

The photographs of the Ta_6_ films show the drastic change of color from brown to deep green when the [{Ta_6_Br^i^_12_}Br^a^_6_]^2−^ (VEC = 14) units are completely reduced to the [{Ta_6_Br^i^_12_}Br^a^_6_]^4−^ (VEC = 16) form, more noticeable for the 2 and 3 wt% doping and that the embedment of as low as 1 wt% of Ta_6_ cluster building block is enough to obtain very colored films (Figure 2f). 

Optical transmittance of the oxidized and reduced Ta_6_ nanocomposite films has been studied in the UV-Vis-NIR range (250–2500 nm). The evolution of the UV-Vis-NIR absorption bands for oxidized and reduced Ta_x@PMMA nanocomposite films are shown in Figure 4a–c. The brown Ta_6__x@PMMA_brown_ spectra present a wide range of up to 90% transmission (500–850 nm) accounting for the good dispersion of the Ta_6_ cluster building blocks in the PMMA matrix. A characteristic absorption is observable in the NIR region at 940 nm that is in accordance with a full oxidized form [{Ta_6_Br^i^_12_}Br^a^_6_]^2−^ (VEC = 14) [28,51]. This band becomes more intense as the cluster content increases (Figure 4d). As described in the literature, oxidized species of [{Ta_6_Br^i^_12_}Br^a^_6_]^2−^ show a characteristic strong band at around 900 nm, accompanied by a shoulder at 750 nm [34,38,51]. Thus, the lack of shoulder at 750 nm, confirms that the cluster oxidation state remains stable during the metathesis reaction, the polymerization and further when the copolymer is solubilized to produce films. The color variation in the films, when DMAP was added to the polymer solution before casting, is accompanied by important changes in the UV-Vis-NIR spectra since intense absorption bands in the visible region at 450, 680 and 740 nm characteristic of green [{Ta_6_Br^i^_12_}Br^a^_6_]^4−^ appeared. These bands have low intensity for the 1 wt% composite and increase strongly for 2 wt% and 3 wt% films; however, the intensity of these bands for 2 and 3 wt% does not increase proportionally to the cluster content as both spectra seem to be approximately the same (Figure 4e). This increase intensity in absorption bands is toward the detriment of the transparency of films whose transmittance at 550 nm decreases from almost 90% for the 1 wt% film to 60% for 2 wt% and 50% for 3 wt%.

The figure of merits (FOM) values, *T*_vis,_
*T*_sol_ and their ratio *T*_vis_/*T*_sol_ allows the evaluation of the efficiency of materials for energy-saving applications. *T*_vis_ accounts for the transmittance of visible light while *T*_sol_ is the transmittance of the total solar radiation [47,59]. The FOM values for the Ta_6_@PMMA composites have been calculated according to the literature and are gathered in Table 2, while the corresponding L, a*, b* color coordinates and CIE diagram is reported in Appendix A. For an ideal blocker that absorbs all the UV (250−400 nm) and NIR (780−2500 nm) and transmits all visible light, the ratio *T*_vis_/*T*_sol_ will be 1.85. However, usually, the values reported in the literature are close to 1 [47,59,60]. In this work, the best values are obtained for composites with 1 wt% loading (1.0 and 1.05). Higher Ta_6_ loading leads to a reduction in FOM which is attributed to a reduction in light transmittance (*T*_vis_) in the visible region, which is undesirable for glazing applications.

It is worth noting that all composites except Ta_3@PMMA_green_ show a calculated *T*_vis_ above 50%, which is the lower limit for window applications [47,59]. At this point, we want to stress that this is the first example of copolymerization of the Ta_6_ cluster building block and whose color is tuned inside the copolymer by controlling the oxidation state. Nevertheless, research on the composition of cluster building blocks and on the film shaping needs to be further conducted in order to optimize the final film properties. The CIE coordinates (Appendix A) show the color coordinates for brown and green nanocomposites. For brown composites, the color darkens as the amount of Ta_6_ cluster increases, while the green coordinates confirm the reduction in Ta_6_ clusters from VEC = 14 to VEC = 16. For VEC = 16 the green color does not evolve greatly between 2 and 3 wt% Ta_6_ cluster loading.

This work opens the doors to the homogeneous embedment of edge-bridged [{Nb_6-x_Ta_x_}X^i^_12_X^a^_6_]^n−^ building blocks (X^i^ = Cl, Br; 0 ≤ x ≤ 6, n = 2, 3, 4) into a wide range of polymers with improved processability. 

## 4. Discussion

The very rich properties of solid-state compounds based on [{Ta_6_Br^i^_12_}Br^a^_6_]^n−^ cluster building blocks and in particular, their absorption and redox properties are very interesting to be used as UV and NIR blockers and/or pigments to tune the hues of materials [28,48,61]. Indeed, {Ta_6_Br^i^_12_}^2+^ core can be oxidized to {Ta_6_Br^i^_12_}^3+^ and {Ta_6_Br^i^_12_}^4+^ reversibly, with a specific absorption behavior for each oxidation state with colors going from green-emerald to red-brownish [28,49,50,51]. The bottleneck in the use of such solid-state compounds is to find straightforward pathways for shaping. The presence of halogens is not suitable for their deposition onto surfaces by techniques such as pulse laser deposition or magnetron. Compared to the majority of ceramics, many cluster compounds are soluble in common organic solvents allowing the use of low-cost shaping in solution. As illustrated here, the possibility to dissolve K_4_[{Ta_6_Br^i^_12_}Br^a^_6_] enables this solid-state cluster to meet the chemistry of PMMA, as a widespread polymer used in industry. Therefore, the introduction of such cluster core into PMMA offers the opportunity to develop multifunctional composite coatings for which the absorption properties can be tuned. Such hybrid composites will take advantage of the synergy between individual UV/ NIR blocking properties of inorganic compounds and the mechanical and optical transparency of polymer matrices. PMMA, also known as acrylic glass, is a versatile polymer that shows excellent light transmission (from 360 to 1000 nm), resistance to weathering, robustness in outdoor conditions and very low thermal conductivity, which makes it an excellent candidate as a host material for thermal-control applications. One strategy to embed inorganic UV/NIR blocking particles into PMMA for solar control applications is by a direct solvent dispersion [2]. Generally, this technique results in non-homogeneous, translucent films due to polarity and a refractive index mismatch between the particles and the polymer matrix. Homogenous and transparent films have been obtained by in situ polymerizations of nanoparticles/MMA monomer dispersions [15,62]. However, the lack of interactions between both components could lead to the leakage of particles into the environment. To overcome these problems in hybrid materials, inorganic counterparts can be covalently bounded to the polymer matrix. For instance, UV- NIR blocking particles such as ZnO, or silver nanoprisms have been successfully introduced into poly(methyl methacrylate) (PMMA) matrix [46,60].

Inorganic UV absorbers can be embedded into the PMMA matrix to improve its resistance to UV photodegradation, which occurs by irradiation between 260 and 300 nm [63,64]. Typically, wide band-gap metal oxide nanoparticles such as ZnO, TiO_2_, CeO_2_ and In_2_O_3_ have been used to improve weather conditions of PMMA [4,5,65,66]. These absorbers are capable to absorb radiation down to 350 nm, however, to avoid diffusion and improve transmittance in the visible range particles should be smaller than 40 nm to lead to homogeneous composites and also shape has to be carefully controlled during synthesis [65]. 

For UV shielding purposes, green Ta_6_ films show enhanced UV absorbance than the corresponding brown ones since Ta_6__2@PMMA_green_ and Ta_6__3@PMMA_geen_ having 2 and 3 wt%, respectively block entirely UV radiation up to 400 nm while Ta_6__1@PMMA_green_ blocks up to 366 nm. Brown films, on the contrary, show a transmittance window between 300 and 375 nm whose maximum decreases with increasing Ta_6_ cluster content, from 66% for 1 wt% content to 16% for 3 wt% content. Thus increasing the amount of Ta_6_ embedded may lead to an increase in the blocking properties of the UV radiation. 

Among the NIR radiation (700−2500 nm), the short wavelengths (700−1100 nm) are responsible for heat and thus they should be blocked for NIR shielding applications [67]. This can be achieved by adding to the glass a coating containing inorganic compounds capable of either reflecting or absorbing NIR radiation. Some inorganic reflectors have been embedded into the PMMA matrix for these purposes such as ZnO, SnO_2_-Al_2_O_3_ or Cs_0,32_WO_3_ tungsten bronzes [2,15,68]. Reflectance between 40−50% can be achieved for ZnO nanocomposites; however, in this study, the FOM values have not been reported. The transmission of solar radiation T_sol_ can be effectively blocked up to 70% for SnO_2_-Al_2_O_3_ paraffin/PMMA and Cs_0,32_WO_3_@PMMA nanocomposites, and a high T_vis_/T_sol_ value of 1.74 has been reported. Research on NIR inorganic absorbers has raised interest in recent years. Inorganic absorbers will induce conduction and convection of half of the heat absorbed inwards and thus, they will be less efficient than reflective coatings. However, limitations on the amount of light that can be reflected are now legislated in some cities [59]. Compared to reflective coatings, reports on coatings embedding inorganic absorbers are quite limited. The T_vis_/T_sol_ values of the Ta_6_@PMMA nanocomposites described in this work are low, the results obtained for 1 and 2 wt% composites are nevertheless comparable to those reported for silver nanoprisms embedded into PMMA matrix and gold nanorod–polyvinylalcohol nanocomposites [59,60]. One of the most studied NIR absorbers is LaB_6_ in the form of particles. The latter has been embedded, for example, in a polyvinyl butyral matrix showing a high T_vis_/T_sol_ of 1.9 for 0.030% LaB_6_ doping [67]. Recently, Lebastard et al. have shown an improved performance for mixed Ta and Nb octahedral metal atom clusters [{Nb_5_TaX^i^_12_}X^a^_6_]^4−^ embedded in polyvinyl alcohol by simple dispersion matrix with T_vis_/T_sol_ of 1.33 that can be further improved to 1.73 when coated on ITO glass [17]. Thus, the PMMA coatings described in this manuscript can be further extended to these new systems with improved performance. 

In recent years, there is an increasing interest in inorganic pigments with various hues that can absorb or reflect NIR light with high transmission in visible light. A series of ion-substituted transition metal pigments with formula Zn_1-x_A_X_WO_4_ (A = Co, Mn, Fe) and Cr-doped BiO4 showing green or brown hues have been reported to induce a high reflectance from 85 to 95% [69,70]. Inorganic pigments as NIR absorbers have been much less reported. One example concerns brown Cu aluminate spinel oxides with broad NIR absorption in the NIR region. However, the FOM values of these compounds were not reported. There is a lack of examples dealing with the embedment by strong interactions into PMMA of pigments with selective UV-NIR absorption. Thus, the development of such coatings showing stable green and brown hues that can be tuned by the oxidation state is of high interest for solar control applications. 

## 5. Conclusions

The work presented herein shows an original strategy to design cluster-based composites combining the advantage of the PMMA matrix (mechanical properties, transparency in the visible region, no yellowing upon aging, and robustness in outdoor conditions) with the UV-NIR blocking properties of metal atom clusters. The method is based on the exchange of the alkali cations counterbalancing the charge of the cluster building blocks in solid-state precursors K_4_[{Ta_6_Br^i^_12_}Br^a^_6_] with organic cations bearing polymerizable moieties. As a result, the anionic cluster building blocks electrostatically interact with the organic cations copolymerized with MMA monomers. This strategy prevents leakage and aggregation of the inorganic moieties, allowing improved processability. Transparent free-standing films have been obtained from pellets, which are readily soluble in all common organic solvents maintaining the integrity of the cluster building block and the copolymer. Here, we show that even though inorganic cluster building blocks are embedded within the PMMA matrix, their optical properties can be efficiently chemically modulated by interaction with DMAP. 

The brown films Ta_6_@PMMA_brown_ contain the Ta_6_ building block in the highest oxidized form (VEC = 14) and can be readily reduced by the addition of two equivalents of DMAP giving rise to green and Ta@PMMA_green_ copolymer films. The highest T_vis_/T_sol_ values are reached for 1 wt% Ta_6_ doping. The UV-Vis-NIR absorption properties can be tuned by playing on the oxidation state of the cluster building blocks. 

Ta_6_@PMMA_green_ films show little absorbance in the red region, contrary to the Ta_6_@PMMA_brown_-shaped films. The latter films are characterized by a large absorption band in the NIR region with an absorption maximum located at 940 nm. 

The strategy presented here can be now extended to the integration in a PMMA matrix of the whole family of [{Nb_6-x_Ta_x_}X^i^_12_X^a^_6_]^n−^ building blocks (X^i^ = Cl, Br; 0 ≤ x ≤ 6, n = 2, 3, 4) starting from the corresponding K_4_[{Nb_6-x_Ta_x_}X^i^_12_X^a^_6_] solid-state precursors [16]. In particular, it has been recently shown that the [{Nb_5_Ta}Cl^i^_12_Cl^a^_6_]^4−^ building block exhibits high absorption properties suitable for solar control applications which are complementary to that of ITO [65]. Indeed, once integrated into polyvinylpyrrolidone and deposited on ITO, the resulting glazing allows the filtering of the most energetic UV and NIR wavelengths. The drawback of such composites based is that the PVP matrix is water soluble and that it exhibits poor mechanical properties.

The integration of [{Nb_6-x_Ta_x_}X^i^_12_X^a^_6_]^n−^ building blocks in the PMMA matrix will enable highly improved possibilities of applications in outdoor conditions. Robust and versatile cluster-based composites with tunable physical properties and in particular tunable hues and redox properties are thus expected. These new composites will bridge the gap between proof of concepts and potential energy-saving applications.

## Figures and Tables

**Figure 1 nanomaterials-13-00144-f001:**
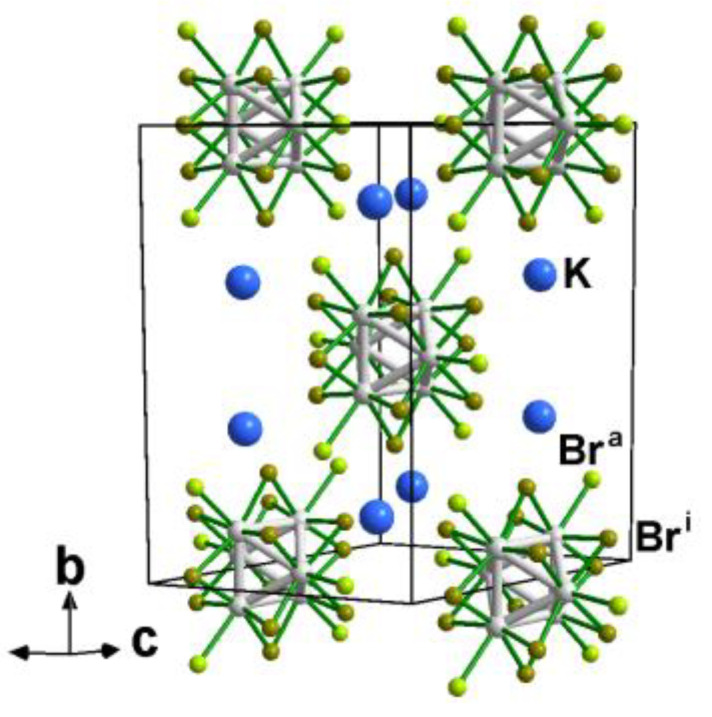
Perspective representation of the unit-cell of K_4_[{Ta_6_Br^i^_12_}Br^a^_6_] evidencing the stacking of the [{Ta_6_Br^i^_12_}Br^a^_6_]^4−^ cluster building blocks and the potassium cations.

**Figure 2 nanomaterials-13-00144-f002:**
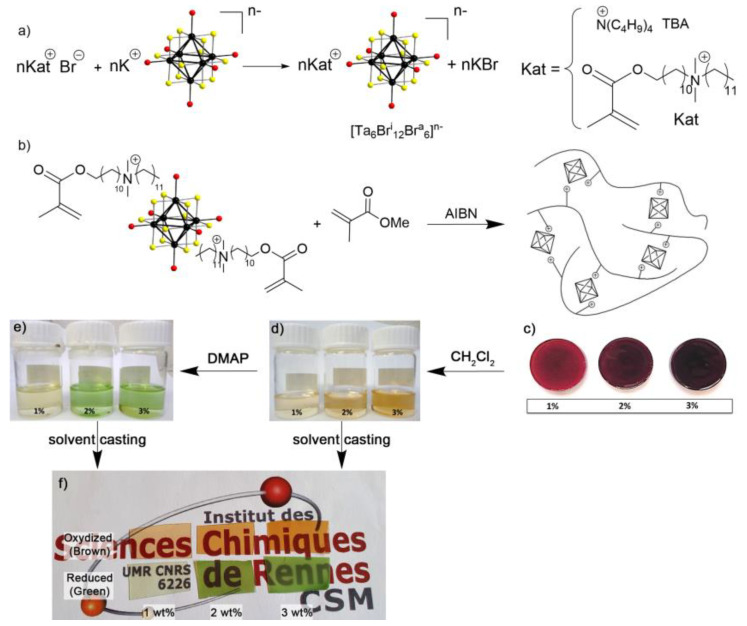
(**a**) Representation of the cationic metathesis reaction between organic ammonium cations (Kat) and potassium in K_4_[{Ta_6_Br^i^_12_}Br^a^_6_], (**b**) Representation of the bulk radical polymerization reaction with MMA carried out at 60 °C for 24 h, (**c**) the brown pellets obtained, (**d**) Solution of brown pellets in CH_2_Cl_2_, (**e**) Reduction in brown Ta_6_ cluster units in solution with two equivalents of DMAP and (**f**) Pictures of brown and green films obtained by solvent casting.

**Figure 3 nanomaterials-13-00144-f003:**
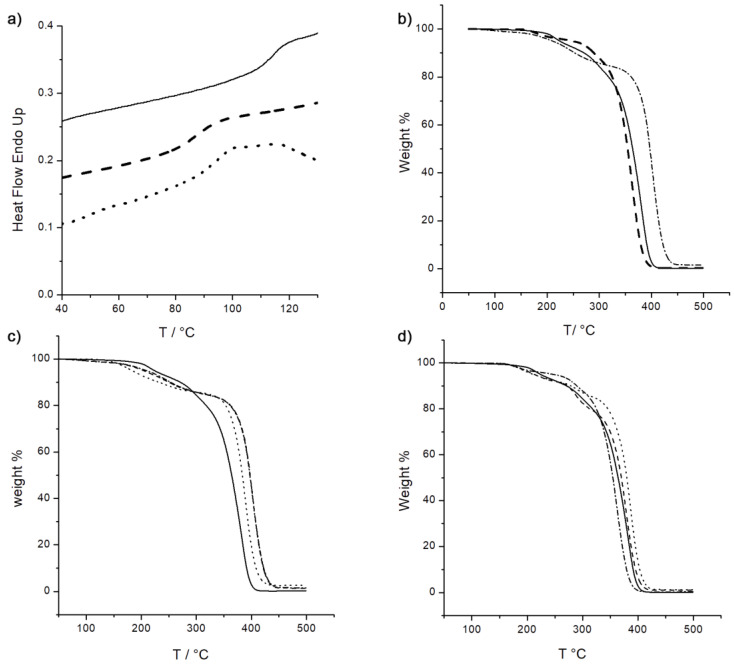
Thermal behavior of PMMA and PMMA composited loaded with 3% of Ta_6_ cluster unit. (**a**) DSC thermograms of PMMA (solid), Ta_3@PMMA_brown_ (dash) and Ta_3@PMMA_green_ (dot), (**b**) TGA curves of PMMA (solid), Ta_3@PMMA_brown_ (dash) and Ta_3@PMMA_green_ (dash-dot), (**c**) TGA of reduced Ta_x@PMMA_green_ of PMMA (solid), Ta_1@PMMA_green_ (dash), Ta_2@PMMA_green_ dot and Ta_3@PMMA_green_ (dash-dot) and (**d**) TGA of oxidized Ta_x@PMMA_brown_ of PMMA (solid), Ta_1@PMMA_brown_ (dash), Ta_2@PMMA_brown_ dot and Ta_3@PMMA_brown_ (dash-dot).

**Figure 4 nanomaterials-13-00144-f004:**
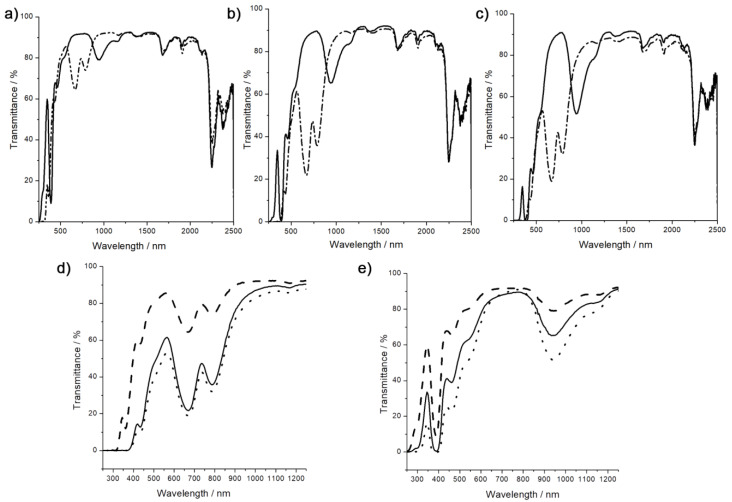
UV-vis-NIR spectra of Ta_x@PMMA nanocomposites in the oxidized and reduced forms. (**a**) Ta_1@PMMA_brown_ (solid) and Ta_1@PMMA_green_ (dash-dot), (**b**) Ta_2@PMMA_brown_ (solid) and Ta_2@PMMA_green_ (dash-dot), (**c**) Ta_3@PMMA_brown_ (solid) and Ta_3@PMMA_green_ (dash-dot), (**c**) and comparison of Ta_x@PMMA films in (**d**) green films at 1 wt% (dash), 2 wt% (solid) and 3 wt% (dot) and (**e**) brown films at 1 wt% (dash), 2 wt% (solid) and 3 wt% (dot).

**Table 1 nanomaterials-13-00144-t001:** Amounts of Ta_6_ cluster in the polymer and amounts of DMAP used for reduction.

wt% Ta_6_ Cluster	Mass Cluster/mg	mmol Cluster	Mass DMAP/mg	mmol DMAP
1	10	0.003	0.006	0.71
2	20	0.006	0.12	1.46
3	30	0.009	0.18	2.20

**Table 2 nanomaterials-13-00144-t002:** Thermal behavior of films by DSC and FOM values of Ta_6_@PMMA nanocomposites.

Sample	*T*_g_ (°C)	*T* _vis_	*T* _sol_	*T* _vis/sol_
PMMA	114			
Ta_6__1@PMMA_brown_	85	81.2	80.8	1.00
Ta_6__2@PMMA_brown_	87	65.3	70.6	0.92
Ta_6__3@PMMA_brown_	89	55.2	64.8	0.85
Ta_6__1@PMMA_green_	79	81.0	77.4	1.05
Ta_6__2@PMMA_green_	93	50.6	53.1	0.95
Ta_6__3@PMMA_green_	97	43.3	49.0	0.88

## Data Availability

Not applicable.

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
