# Peer review of "From Solid-State Cluster Compounds to Functional PMMA-Based Composites with UV and NIR Blocking Properties, and Tuned Hues"

_nanomaterials, 2022, doi:10.3390/nano13010144_

Round 1
Reviewer 1 Report
Peer review for manuscript: “From solid state cluster compounds to functional PMMA-based composites with UV and NIR blocking properties, and tuned hues”
The under-review manuscript is a successful attempt of synthesizing Ta6 cluster/PMMA composites for energy saving application in buildings by blocking UV and NIR rays. The text is generally well written and includes many literature citations, indicating that the authors performed an adequate literature review related to their work. However, since it is a complicated work with a lot of information an effort should be made to revise it to a more comprehensive manuscript with established arguments. In my opinion the comments that follow will help towards that direction. The results presented here are very interesting since this is the first example of copolymerization of Ta6 cluster building block and whose color is tuned inside the copolymer by controlling the oxidation state.
Remarks
Comment 1. The authors should improve the grammar and syntax e.g. lines 27, 31, 38, 193.
Comment 2. In the text, many abbreviations are used. Thus, the authors should add a list/table of abbreviations to facilitate the reader.
Comment 3. The authors should provide more details for the employed reagents (e.g. supplier, CAS No) (Line 110).
Comment 4. The preparation process of the PMMA composites involves a few steps. Thus, a graphic depiction of the preparation process would shed light on them.
Comment 5. The authors should provide more details regarding the 1H-NMR measurement (Line 160)
Comment 6. The authors should rephrase very long sentences. For example, in lines 209-215 the whole paragraph is one sentence.
Comment 7. Photos of the green and brown pellets would be a nice addition to the text.
Comment 8. “Brown homogeneous pellets denominated Ta_1@PMMAbrown, Ta_2@PMMAbrown and Ta_3@PMMA brown (containing 1, 2 and 3 wt% Ta6 cluster, respectively) were obtained.” (Line 262-263). How the authors determined the dispersion degree of the Ta6 cluster in the PMMA matrix? Since the homogeneous dispersion of the cluster is important for the properties of the final composite the claimed homogeneity of 1, 2 and 3 wt% composites should be determined and presented.
Comment 9. In one of the initial preparation steps the brown and the green pellets are synthesized. Photos of the prepared pellets should be added.
Comment 10. The authors should add details on the UV-Vis-NIR experiments. They are the most important experiments since they are the only proof of the composites UV and NIR blocking properties they should be thoroughly described. The detailed description includes both the experimental and the results segment.
Comment 11. Figure 3: Why only the 3wt% Ta6 curves are presented? Why Ta6_3@PMMA green sample is not included in the TGA graph? The conclusion in lines 325-326 is only justified by the changes in the Tg?
The addition of Ta6 cluster has difference influence on the thermal properties of the resulting composite, depending upon the initial cluster concentration. What changes between 1% and 2% cluster addition? The authors should discuss it in accordance with the relevant literature in order to interpret it and extract useful conclusions.
Comment 12. Line 373: The reference “Figure 4a: should be added to guide the reader.
Comment 13. Lines 392-394: The authors mention that the bands intensity is low for 1wt% composite and strongly increases for 2 and wt%. In the next sentence it is mentioned that the bands intensity remains for 1, 2 and 3wt% cluster content. The segment requires revision and clarification.
Comment 14. The authors should complete the sentence in line 396.
Comment 15. Figure 4
· The picture 4a should be replaced with a better one. The text is not easily spotted, and it is not suitable for scientific manuscript. Labels for oxidized and reduced should also be included on the photo.
· In the picture 4a the sample Ta_3@PMMA green is less transparent compared to its brown counterpart. The authors should explain it. If the transparence depends on homogeneous dispersion of the cluster in the polymeric matrix, then the 3wt% films (green and brown) shouldn’t have the same transparence degree?
· The authors should add legend description for figures 4c and 4d.
Comment 16. Line 441: the table’s number should be changed from 1 to 2. The necessary changes should be performed in the text as well.
Comment 17. Lines 495-496: The authors should mention here that increase of Ta6 loading also leads to Tvis decrease resulting in increased film opacity which is undesirable effect. Thus, further research/ optimization is required to be certain on the conditions that improve the films properties. The conclusion “Thus increasing the amount of Ta6 embedded may lead to block completely the UV radiation.” is inadequate.
Comment 18. The authors assume that the mechanical properties of the PMMA have remained intact after the Ta6 cluster addition. Can this be justified by the literature?
Comment 19. Lines 555-557 are not conclusions. The authors should add conclusions that correlate to the mentioned results, or they should remove them.
Comment 20. Line 754: The number 1 should be removed.
Comment 21. In several parts , the authors refer to supporting info, where is rather complicating to compare figures from the supporting info to the ones appearing in the manuscript. Probably, re-arrangement of the Figures should take place.
Comment 22. The authors should use one common way of righting the range of the wavelength from higher to lower wavelength.
Comment 23. The authors should describe in each analytical method they used, why they used it and what they were aiming to. It is more comprehensive for the people to read it to keep up with the storyline of the manuscript.

Author Response
Please see the attachement

Reviewer 2 Report
The manucript by Amelia-Cortes reports new nanocomposite materials with UV-NIR blocking properties and hues ranging from green to brown, which were prepared by integrating {Ta6Br12} clusters in a PMMA matrix. I feel that the matter reported in the manuscript is worth publishing after attending several issues. These are following:
1. The authors report optical properties of the films for two extreme oxidation states of the cluster: +2 and +4. What about +3? It should be accessible by choosing the correct amount of reductant. Such study will enhance the scope of the paper
2. What about self-reduction of the 2+ films after longer irradiation time (days) (and vice versa, self-oxidation of the 4+ films)? This process may well give the 3+ state.
3. I feel uneasy with the explanation of the role of dimethylaminopyridine as reducing agent as is given in th text. Two equivalents were taken, but the product, apparently, is simply protonation product. Now, protonation is not oxidation. Where does the proton comes? Dimethylaminopyridine (DMAP) contains a NMe2 group which can act as reductant, finally loosing a CH3 group. It means that, if we want intact DMAP to be protonated and balance the charge, we need at least one more equivalent to be consumed as reductant.
4. Recently a paper appeared on {Ta6Br12}-diven photocatalysis: J.S. Hernandez, M.S. Shamshurin, M. Puche, M.N. Sokolov, M. Feliz, Nanomaterials, 2022, 12, 3647
Author Response
Please see the attachement

Round 2
Reviewer 1 Report
Dear authors,
I wish to thank you for taking into consideration the comments and the suggestions made to the current manuscript.
No further changes are requested.
The manuscript is considered accepted.
Reviewer 2 Report
I am satisfied with the new version. Still a bit uneasy about DMAP reduction but I agree that the reduction of the cluster by DMAP is the matter of fact and exact details are not relevant for the purpose of this paper.